# Diffusive tail anchorage determines velocity and force produced by kinesin-14 between crosslinked microtubules

Annemarie Lüdecke[1], Anja-Maria Seidel[1], Marcus Braun [1,2], Zdenek Lansky [1,2] & Stefan Diez [1,3]

Form and function of the mitotic spindle depend on motor proteins that crosslink microtubules and move them relative to each other. Among these are kinesin-14s, such as Ncd, which interact with one microtubule via their non-processive motor domains and with another via their diffusive tail domains, the latter allowing the protein to slip along the microtubule surface. Little is known about the influence of the tail domains on the protein's performance. Here, we show that diffusive anchorage of Ncd's tail domains impacts velocity and force considerably. Tail domain slippage reduced velocities from 270 nm s$^{-1}$ to 60 nm s$^{-1}$ and forces from several piconewtons to the sub-piconewton range. These findings challenge the notion that kinesin-14 may act as an antagonizer of other crosslinking motors, such as kinesin-5, during mitosis. It rather suggests a role of kinesin-14 as a flexible element, pliantly sliding and crosslinking microtubules to facilitate remodeling of the mitotic spindle.

[1] B CUBE–Center for Molecular Bioengineering, Technische Universität Dresden, Arnoldstr. 18, 01307 Dresden, Germany. [2] Institute of Biotechnology of the Czech Academy of Sciences, BIOCEV, Prumyslova 595, 252 50 Vestec, Czech Republic. [3] Max Planck Institute of Molecular Cell Biology and Genetics, Pfotenhauerstraße 108, 01307 Dresden, Germany. Correspondence and requests for materials should be addressed to M.B. (email: marcus.braun@ibt.cas.cz) or to Z.L. (email: zdenek.lansky@ibt.cas.cz) or to S.D. (email: stefan.diez@tu-dresden.de)

The cellular function and molecular mechanism of mitotic, microtubule-crosslinking kinesins have been studied by genetically perturbing whole cells[1–6] as well as by performing in vitro reconstitution assays[7–13]. For example, it is known that tetrameric kinesin-5 motors step processively on both microtubules that they crosslink[10] and that their force output grows linearly with the number of engaged motor proteins, with increases over 1 pN per motor protein[12]. Less is known about the molecular mechanism of dimeric crosslinking motor proteins, like kinesin-14, which bind to one microtubule diffusively via their non-motor tail domains and step on another microtubule non-processively with their motor domains[8]. So far, in vitro studies have mainly concentrated on the kinesin-14 motor domain, its mechano-chemical cycle[7,13], effects of cooperativity[9] and off-axis power stroke[7,11]. Only a few studies[8,14] have attempted to reconstitute the native function of kinesin-14, namely to crosslink and slide microtubules. Thus, it remains unclear how kinesin-14s fulfill their biological function, e.g. how much force they can exert in between crosslinked microtubules. As it has recently been shown that the diffusive anchorage of kinesin-1 in a lipid bilayer can regulate the transport efficiency to great effect[15], we wondered if a similar mechanism might be at play with kinesin-14, such that velocity and force produced would be determined by the tail domain-mediated diffusive anchorage of the motors on the microtubules. Here, we show that the diffusive anchorage indeed impacts the force that kinesin-14 motors can generate between two microtubules and we argue that modifications of the anchorage interfaces may act as a regulatory factor influencing the force generation by kinesin-14.

## Results

**Statically anchored kinesin-14 motors exert pN-range forces.** To understand the influence of the diffusive anchorage of kinesin-14 we performed in vitro experiments using the drosophila kinesin-14, non-claret disjunctional (Ncd). First, we studied the velocity and force produced by teams of statically anchored Ncd motor domains (gliding geometry, Fig. 1a). To that end, we immobilized truncated, tail-less, SNAP$_f$-tagged, dimeric Ncd-motor constructs (NcdΔtail) on a glass surface using anti-SNAP-tag antibodies. Fluorescently labeled, biotinylated microtubules were added to the flow chamber in the presence of 1 mM adenosine triphosphate (ATP) and observed by fluorescence microscopy (Supplementary Movie 1). Microtubules showed gliding motility with velocities of $269 \pm 95$ nm s$^{-1}$ (mean ± SD, $N = 879$ microtubules, Fig. 1b), independent of microtubule length and thus independent of motor number (Fig. 1c, Supplementary Figure 1a). To quantify the generated forces, NeutrAvidin-coated silica microspheres were attached to gliding microtubules using optical tweezers. After binding to the microtubules, the microspheres were transported out of the center of the optical tweezers, and force and displacement signals were recorded (Fig. 1d). The forces exerted on the microspheres were calculated from the force data by averaging over suitable time frames (see Methods). Most microtubules were stopped in their movement once a sufficient counteracting force was reached ($N = 78$ microtubules). These stopping forces (occasionally exceeding 10 pN, Fig. 1e) scaled with the number of Ncd motors, calculated from microtubule length and motor surface density (see Methods, Fig. 1f, Supplementary Figure 1b), as had been suggested previously[9]. In some cases ($N = 23$ microtubules), the microtubule–microsphere attachments broke before the microtubules halted or the microspheres left the linear regime of the optical tweezers, as the force exerted by the optical tweezers was insufficient to stop the forward microtubule movement. Forces measured in these events often exceeded the forces measured on stopped

microtubules and occurred predominately for long microtubules (>6 µm). We quantified forces up to 27 pN exerted on Ncd-driven microtubules. Thus, teams of surface-immobilized NcdΔtail motors can exert high forces and exhibit an intrinsic stopping force that is comparable to teams of other kinesin motors[12,16–18].

**Diffusively anchored kinesin-14 motors exert sub-pN forces.** Second, we investigated how much of the generated force was effectively transmitted between microtubules in the biologically relevant context of the sliding of antiparallel microtubule pairs by full-length Ncd (flNcd) motors. To that end, we devised an in vitro assay in which individual microtubules were surface-immobilized and crosslinked by green fluorescent protein (GFP)-tagged flNcd to individual transported microtubules (sliding geometry, Fig. 2a, Supplementary Movie 2, note that in this assay geometry only antiparallel microtubules exhibit sliding motion, whereas parallel microtubules are statically crosslinked[8]). We measured sliding velocities of $57 \pm 33$ nm s$^{-1}$ (mean ± SD, $N = 265$ microtubules, Fig. 2b) in the presence of 1 mM ATP, independent of microtubule length and thus motor number (Fig. 2c, Supplementary Figure 2). As in the previous setup, we determined the stopping forces by attaching silica microspheres to the transported microtubules using optical tweezers (Fig. 2a). To our surprise, the stopping forces did not surpass noise levels (Fig. 2d). To nevertheless quantify the forces, we conducted force-velocity measurements. We applied a constant force between −5 pN and +5 pN using the force-feedback routine provided by the instrument's manufacturer and recorded the microscope stage velocity necessary to apply the desired force using stationary optical tweezers. Maximum forces of between 2 pN and 5 pN corresponded to microscope stage velocities of around 200–1000 nm s$^{-1}$ for different microtubule pairs. For all microtubule pairs, the relationship of force and velocity was linear (Fig. 2e). To calculate the slope of the force-velocity curve, we used a linear regression and weighted the data points with a factor representing the steadiness of microtubule–microtubule sliding. The combination of the slope of the force-velocity plot, i.e., the friction introduced by the flNcd proteins, and the observed, unloaded velocity of every transported microtubule allowed us to calculate the stopping force (see Methods and Fig. 2e). The median stopping-force measured on $N = 33$ independent microtubule pairs of microtubules was below 1 pN (Fig. 2f). Possibly due to the small force magnitudes, we found no significant correlation between stopping force and motor number, the latter of which was simultaneously quantified using total-internal reflection fluorescence microscopy (see Methods).

We independently quantified the stopping force produced by flNcd in two additional ways. First, we re-evaluated data published previously[19], in which microtubules were crosslinked simultaneously by flNcd and Ase1, a passive microtubule crosslinker confined to microtubule overlaps. In shortening microtubule overlaps, the constant number of diffusive, confined Ase1 molecules exerted entropic expansion forces which are proportional to the number of Ase1 molecules and inversely proportional to the overlap length[20]. In contrast, flNcd did not enrich in the microtubule overlaps, consequently sliding microtubules apart only until the opposing entropic Ase1 forces balanced the sliding forces[19,20]. We quantified the forces generated by teams of Ncd in 49 microtubule pairs by quantifying the entropic forces of Ase1 when sliding had halted (Fig. 2g, see Methods). The force distribution agrees well with the force distribution measured by optical tweezers (Fig. 2f). Second, we performed microtubule buckling assays. In these assays, microtubule buckling is associated with a force, which can be estimated from the known

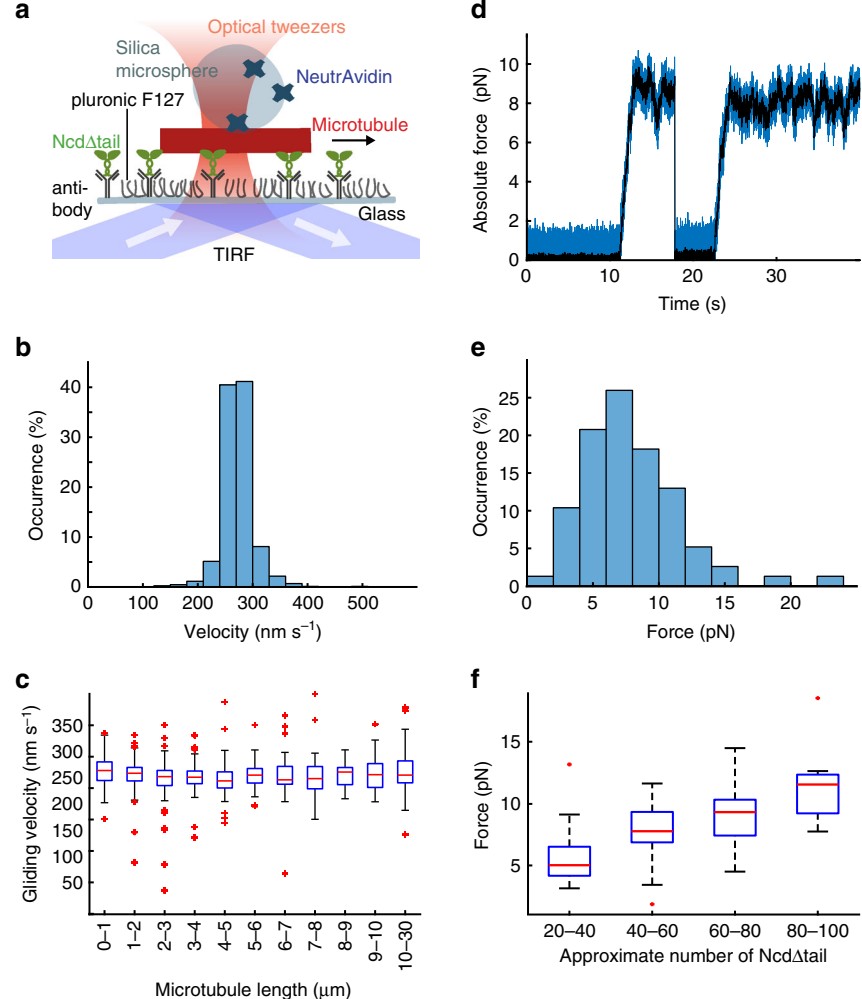

**Fig. 1** Statically anchored kinesin-14 motors exert pN-range forces. **a** Truncated, SNAP$_f$-tagged Ncd (NcdΔtail) motors were surface immobilized by anti-SNAP-antibodies. The surface was passivated using the polymer pluronic F127. Biotinylated microtubules interacted with NcdΔtail motors (gliding geometry) and with silica microspheres that were manipulated by optical tweezers. **b** Velocity distribution of all gliding microtubules ($N = 867$). **c** Microtubule gliding velocities without external load were constant over a wide range of microtubule lengths, corresponding to a wide range of NcdΔtail motor numbers. The values represent averaged velocity values per microtubule, which were afterwards time weighted (this data treatment was chosen for better comparison to Fig. 2c). The number of microtubules per depicted bin is shown in Supplementary Figure 1 and ranges from 25 to 124 microtubules. For the boxplot elements description see Methods. **d** Characteristic force trace for a microsphere attached to a microtubule that was transported out of the center of the optical tweezers until the stopping force was reached. After that the bead detached and rebound to the same microtubule and was again transported until the stopping force was reached. The stopping force was calculated by averaging over the time period when the microtubule was stopped (see Methods). Note that absolute force is plotted here (see Methods). **e** Distribution of forces exerted on gliding microtubules by teams of NcdΔtail immobilized to a glass surface ($N = 78$). **f** Microtubule stopping force, binned according to the approximate number of NcdΔtail motors. The motor number was calculated from the microtubule length and the motor density on the surface, which was quantified by fluorescence intensity (see Methods). For the boxplot elements description see Methods

flexural rigidity of microtubules and the length of the bent part of the microtubule, with shorter microtubule lengths corresponding to higher forces[21,22]. The shortest microtubule segments we observed buckling only buckled after several tens of seconds of thermal fluctuation, suggesting that the force just sufficed to bend the microtubule. These segments were of 8.3 μm and 10 μm lengths, corresponding to 1.1 pN and 0.7 pN respectively (Fig. 2h, i and Supplementary Movie 3). Evaluating these three independent methods of force measurement, we suggest that teams of slippage-prone flNcd motors exert only sub-piconewton forces when sliding antiparallel microtubules.

Thus, in summary, we found that statically anchored Ncd motors (NcdΔtail) support fast microtubule gliding and, collectively, produce high forces, comparable to other kinesin motors. By contrast, native, diffusively anchored, full-length

motors (flNcd) support only slow microtubule sliding and generate only low, sub-piconewton, forces. We thus argue that it is the tail-mediated anchoring of the motor that regulates the transmission of Ncd generated forces.

**Restricting the tail diffusion restores pN-range forces**. If the force transmission depends on the anchoring of the Ncd motors, then restricting the diffusibility of the Ncd tail domains should lead to increased velocities of microtubule transport and the generation of higher forces. To test this prediction, we analyzed assays where transported microtubules bound to one or more surface-immobilized microtubules in a non-aligned manner (Fig. 3a, b, see Supplementary Movie 2 for two examples). In this non-aligned geometry, Ncd tail domains were present, but their

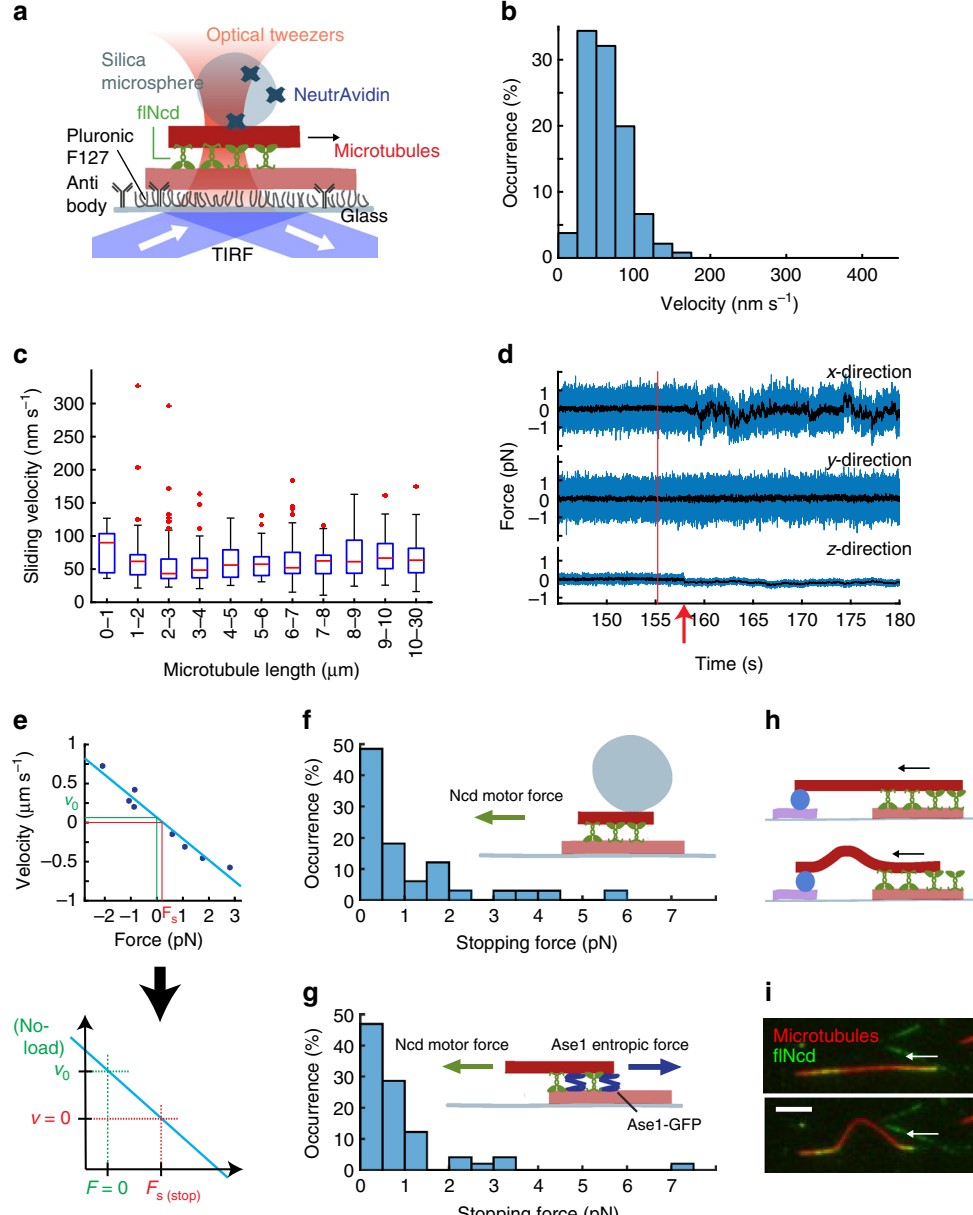

**Fig. 2** Diffusively anchored kinesin-14 motors exert sub-pN forces. **a** Sparsely cy5-labeled, digoxigeninated microtubules were surface immobilized using anti-digoxigenin antibodies. The surface was passivated using the polymer pluronic F127. Full-length Ncd (flNcd) motors crosslinked surface-immobilized microtubules and transported microtubules, which were densely cy5-labeled and biotinylated (sliding assay geometry). A NeutrAvidin-coated microsphere bound to the transported microtubule was manipulated using optical tweezers. **b** Velocity distribution of microtubules transported by flNcd ($N = 265$). **c** Velocities of transported microtubules without external load were constant over a wide range of microtubule lengths. The number of microtubules per depicted bin is shown in Supplementary Figure 2. For the boxplot elements description see Methods. **d** Characteristic force trace for a microsphere attached to a microtubule that is transported by flNcd. Note that once the microsphere attaches to the microtubule, sliding motion stops immediately and the microsphere is not transported out of the trap center. Direction of microtubule sliding is parallel to the x-axis. The binding event is marked with a red arrow. The last movement of the microscope stage prior to binding is marked with a vertical red line. **e** Typical force-velocity measurement used to quantify the stopping force of flNcd in microtubule overlaps. Below, a schematic zoom-in is shown. **f** Stopping force distribution of 33 independent transported microtubules (for method of force quantification see Fig. 2e). **g** Force distribution of 49 independent microtubule pairs estimated from the microtubule overlap length and the number of Ase1 molecules within that overlap. Inset: Depiction of the assay used to determine this force distribution. The non-enzymatic, diffusive crosslinker Ase1, when confined in a microtubule overlap, exerts an entropic force resisting its compaction in the shortening microtubule overlap. Ncd-driven microtubule sliding stops when force balance is reached[19]. **h** Schematic depiction of the microtubule buckling assay. Two sets of microtubules were surface immobilized using anti-digoxigenin antibodies: dimly rhodamine-labeled digoxigeninated and cy5-labeled, biotinylated, digoxigeninated microtubules. NeutrAvidin was used to coat the biotinylated microtubules. flNcd was added and biotinylated, brightly rhodamine-labeled transport microtubules were allowed to bind. Upon encountering NeutrAvidin the transported microtubule stopped and, if Ncd exerted sufficient force, buckled. **i** Micrographs showing the buckling event with the highest force, which was calculated to be 1.1 pN. Several tens of seconds of thermal fluctuation preceded the buckling; scale bar 5 μm

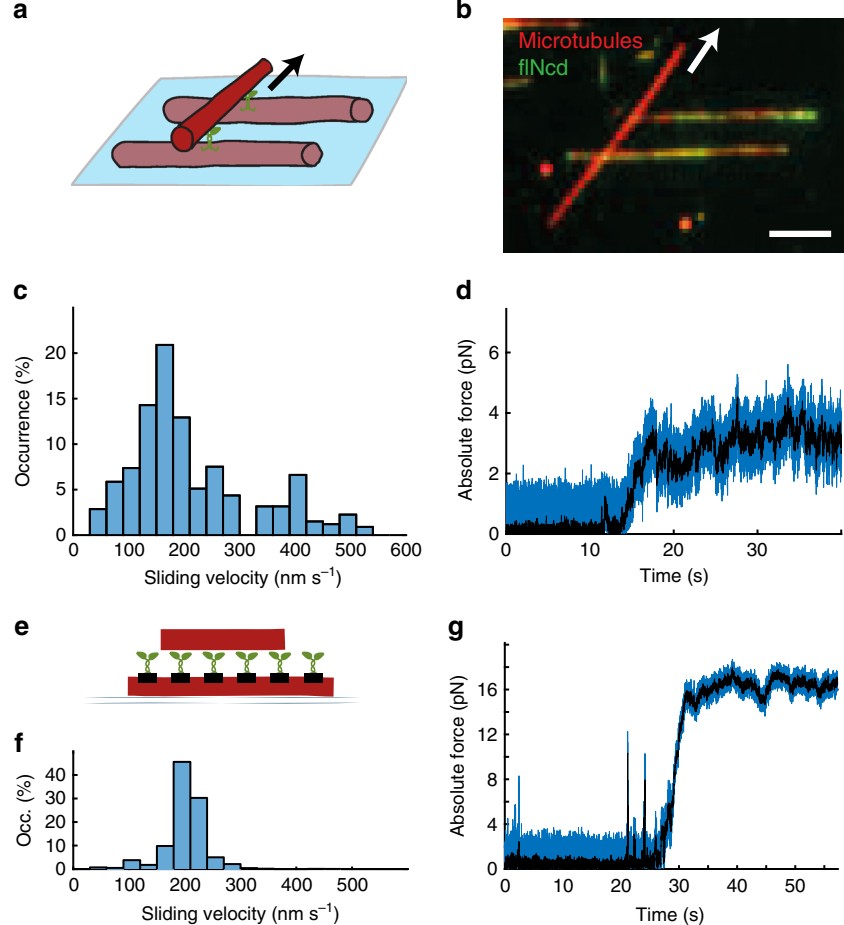

**Fig. 3** Restricting the tail diffusion restores pN-range forces. **a** Depiction of non-aligned microtubules crosslinked by flNcd at an angle. The interaction with two surface-immobilized microtubules prevented the transported microtubule from aligning with either surface-immobilized microtubule. **b** Micrograph of non-aligned crosslinked microtubules; scale bar 5 μm. **c** Velocity distribution of non-aligned microtubules transported by flNcd without external load ($N =$ 83). **d** Characteristic force trace for a microsphere, which was attached to a non-aligned microtubule that was transported by flNcd. The plotted force is absolute force. **e** Depiction of NcdΔtail statically anchored on microtubules, transporting microtubules. **f** Velocity distribution of microtubules transported by NcdΔtail statically anchored on microtubules ($N =$ 200). **g** Characteristic force trace for a microsphere, which was attached to microtubule that was transported by NcdΔtail statically anchored on another microtubules. The plotted force is absolute force

diffusive reach was limited to the short axis of a microtubule, thereby geometrically restricting tail slippage. Furthermore, the numbers of flNcd motors that localized to the intersections to interact with both microtubules were limited. Strikingly, the sliding velocities of non-aligned microtubules were significantly higher than the sliding velocities of aligned microtubules. The median velocity of $180 \pm 130$ nm s$^{-1}$ (median ± SD, $N = 83$) fell in between the velocities of microtubules propelled by fully diffusive flNcd motors and surface-immobilized NcdΔtail motors (Fig. 3c), correlating with the degree of tail diffusibility. We then quantified the force produced by flNcd motors, which crosslinked non-aligned microtubules using optical tweezers. Similar to the measurements on surface-immobilized Ncd motor domains, the stopping forces could be quantified from the raw data in all eight measured events (example in Fig. 3d). We found that the force produced by small teams of flNcd motors with restricted tail diffusion was in the piconewtons range, with the median of $5.9 \pm 2.2$ pN (median ± SD, $N = 8$), easily surpassing the stopping forces for larger groups of diffusively anchored flNcd motors and almost reaching the stopping forces of surface-immobilized NcdΔtail motors. By comparison, the sliding velocities of aligned and non-aligned microtubules were not significantly different for non-diffusive, microtubule-crosslinking kinesin-5

motors (Supplementary Figure 3) for which it was also recently reported that the stopping forces scale linearly with motor number irrespective of the crosslinking geometry[12]. Our results thus show that limiting the diffusibility of the Ncd tail domain by spatial confinement increases the force transmission.

Finally, we asked whether artificially inhibiting the tail diffusibility along the microtubule could generate fast sliding of fully aligned microtubules. To this end, we covalently linked SNAP-tagged NcdΔtail to surface-immobilized microtubules functionalized with benzylguanine (see Methods), and added non-functionalized transported microtubules (Fig. 3e). After specific landing of individual transported microtubules on the immobilized microtubules covered with statically anchored NcdΔtail motors, we observed microtubule sliding at velocities of $205 \pm 37$ nm s$^{-1}$ (mean ± SD, $N = 200$, Fig. 3f), comparable to the velocities observed during the sliding of non-aligned microtubules propelled by flNcd with restricted diffusibility. Force measurements on such sliding microtubules (6 pN and 16 pN, $N = 2$) indicated that forces comparable to the forces generated by surface-immobilized NcdΔtail (example in Fig. 3g) could be generated. Together, these findings show that the anchoring of the kinesin-14 molecules, static or diffusible, determines the maximum force the motors can generate between

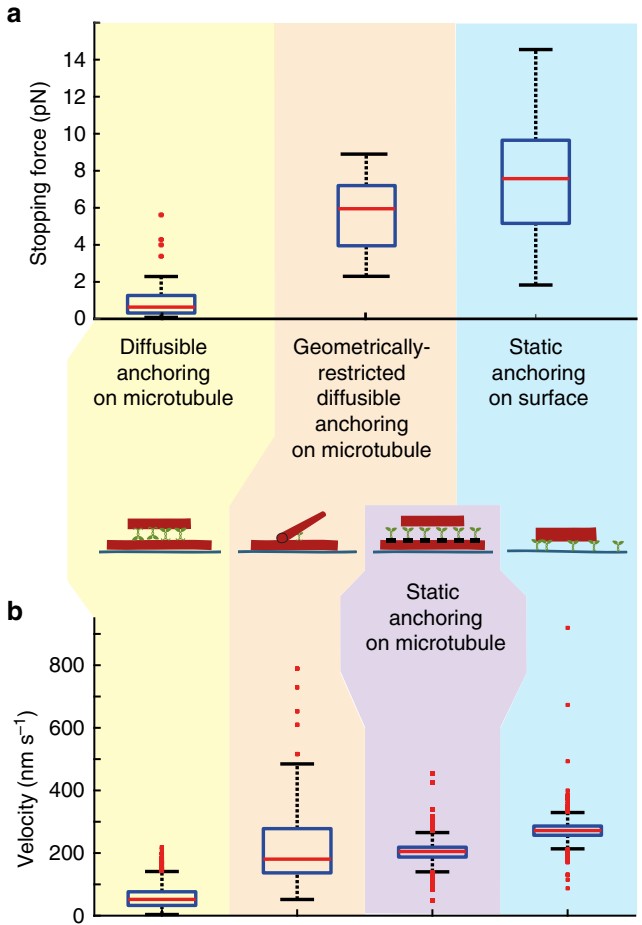

**Fig. 4** Comparison of the different geometries. **a** Comparison of stopping forces for the cases of diffusively anchored flNcd ($N = 33$ microtubules), flNcd with geometrically restricted diffusible anchoring (between non-aligned microtubules, $N = 8$) and NcdΔtail statically anchored on a surface ($N = 78$). Several values above 14 pN lie outside of the depicted range for the case of surface-immobilized motor domains. **b** Comparison of microtubule transport velocities when propelled by diffusively anchored flNcd ($N = 265$), flNcd with geometrically restricted diffusible anchoring (between non-aligned microtubules, $N = 83$), NcdΔtail statically anchored on microtubules ($N = 200$) and NcdΔtail statically anchored on a surface ($N = 879$). Two values lie outside of the depicted range for the cases of geometrically hindered diffusion and one value for the case of surface-immobilized motor domains. For the boxplot elements description see Methods

two microtubules, and thus also the microtubule sliding velocities (Fig. 4).

## Discussion

Our experiments show that diffusibly anchored Ncd motors (such as in the sliding experiments in Fig. 2) transport microtubules slower and generate lower forces than Ncd, which is anchored statically (such as in the gliding experiments in Fig. 1). These results suggest that the anchoring of the tail domain determines the force transmission between the moving microtubules. In the following, we discuss and rule out possible alternative explanations for the observed difference in velocities and forces between sliding and gliding experiments, namely (i) the absence of the Ncd tail domain in the microtubule gliding assays; (ii) collisions between motors moving in opposite directions between sliding

microtubules (that could not occur during microtubule gliding—which is propelled by statically anchored motors), and (iii) a different number or density of engaged motors in the sliding and gliding experiments.

(i) In our NcdΔtail gliding experiments, the velocities and stopping forces were higher than in the flNcd sliding experiments. The presence of the tail domain in the sliding assays might reduce velocity and stopping force possibly due to inhibition of the motor domain activity, as observed for other kinesins[23,24], or by direct, tail-mediated crosslinking of the two microtubules, which would induce additional frictional drag. However, during the flNcd sliding experiments we observed that non-aligned microtubules moved at velocities, and generated stopping forces, significantly higher than aligned microtubules. Because in both cases (aligned and non-aligned microtubule sliding) the tail domain is present, we argue that the absence of the tail domain cannot explain the increased velocities and generated forces during microtubule gliding. We thus argue that motor domains are not inhibited by the presence of the tail domains. Finally, dimeric Ncd tail constructs (lacking the motor domains) do not induce any specific microtubule–microtubule interaction[8,25], ruling out the possibility that crosslinking of microtubules by kinesin-14 tail domains is the cause for the observed reduction in velocity and stopping forces during flNcd-propelled microtubule sliding, compared to NcdΔtail-propelled microtubule gliding.

(ii) During the sliding of antiparallel microtubules, flNcd motors bound to the two respective microtubules move in opposite directions within the overlap[8]. The opposing motions might result in collisions between the motors that might lead to a decrease in the sliding velocity. However, comparing the integrated GFP intensities of the overlap-localized flNcd to the intensity of a single GFP suggests that the density of flNcd in the overlap in our experiments was typically in the order of 1 flNcd molecule per 100 nm, which makes these collisions unlikely. Moreover, in our sliding experiments we never observed the formation of permanent or temporary multi-motor clusters that would indicate that the motors interacted with each other in the microtubule overlap. We thus rule out motor collisions as a cause for the reduced sliding velocities and stopping forces.

(iii) Finally, the velocity of microtubule motion could be affected by the number or the density of engaged motors. It has been reported that increased densities of other kinesin-14 motors decrease the microtubule transport velocity, presumably due to steric effects[14,25]. Analysis of GFP-flNcd fluorescence signal along the microtubule lengths showed that, similarly to the sliding experiments, also during microtubule gliding, the motor densities were in the range of 1 motor per 100 nm of microtubule length. Additionally, we observed that in our gliding experiments, short (<0.5 μm) microtubules often detached from the surface. These observations suggest that the motor density in both gliding and sliding geometry was in the same low range and that mutual steric hindering of the motors that might cause a velocity decrease was rather unlikely. As the distribution of microtubule lengths was the same in both the gliding and sliding experiments, the numbers of engaged motors were also presumably in the same range. Moreover, we did not observe any dependence of the microtubule velocity on the number of motors neither in the gliding nor in the sliding geometry (Figs. 1c and 2c). We thus rule out a different number or density of engaged motors as the cause for the observed differences in velocity and stopping force between the gliding and sliding experiments.

In conclusion, we suggest that it is the diffusive anchoring of the tail domain on one microtubule, which limits the force transmission onto the other sliding microtubule. Teams of diffusively anchored flNcd motors produced forces below 1 pN (sliding crosslinked microtubules at velocities of about 60 nm s$^{-1}$), while

teams of statically anchored NcdΔtail motors generated significantly higher forces in the range of 10 pN and above (at velocities of about 200 nm s$^{-1}$) (Fig. 4). An analogy of these situations could be a person pushing a car while wearing roller skates—the low-force transmission onto the car does not reflect the maximum force produced by the person, but rather the slipping of the roller skates (Supplementary Figure 5). The maximum force that the person can exert is limited by the frictional force of the roller skates according to Newton's third law. This force can, analogously, be increased both by changing into shoes with a stronger grip (representing our assays with statically anchored motors) or turning the roller skates sideways, uncoupling sideways slippage and forward movement (representing our sliding assays with non-aligned microtubules). Both approaches enable the person to exert higher forces onto the car, and thus also increase the velocity of the car.

What role could be played by a low-force microtubule-crosslinking motor in the mitotic spindle? In vivo studies have implicated kinesin-14 in spindle length control[1,5] and spindle pole focusing[2,3]. Kinesin-14's role in spindle length control might be to mediate microtubule nucleation[26] rather than exerting direct pushing or pulling forces against other crosslinking motors such as kinesin-5[6,27–29]. Support for the latter is also provided by previous results from Hentrich and Surrey[14] who showed in vitro that a 100-fold excess of kinesin-14 motors was needed to counteract kinesin-5 motors in between sliding microtubules. Our measurements provide direct evidence supporting their hypothesis that the inefficient force production of kinesin-14 might be a consequence of kinesin-14's tail diffusion on the microtubule surface. Regarding the role of kinesin-14 in sliding microtubules, its primary role might be to slide newly nucleated microtubules that are not yet otherwise tethered[30] to the spindle poles[1], while also separating antiparallel microtubules by sliding and bundling parallel microtubules by crosslinking[8,31]. Both mechanisms do not depend on the exertion of high forces and could explain kinesin-14's role in spindle pole focusing[1,3]. Potentially, spindle pole focusing could also benefit from kinesin-14 exerting higher forces on non-aligned microtubules, thus pushing those efficiently out of the way. It remains to be seen if those are kinesin-14's only functions or whether the force produced by kinesin-14 is regulated by changing tail diffusivity in vivo. Possible mechanisms for that include post-translational modifications of the microtubule surface[32] and/or of the kinesin-14 tail domain[33] as well as docking to auxiliary proteins like EB1[34] that may anchor kinesin-14 firmly to microtubule tips. These mechanisms may increase the velocity and force production of kinesin-14.

## Methods

**Protein purification**. Recombinant histidine-tagged *Drosophila melanogaster* full-length GFP-Ncd (flNcd) and truncated GFP-Snap-Ncd motor domains (a.a. 196–700, NcdΔtail) were expressed in SF9 insect cells using a baculovirus expression system[35.] Protein lysate in lysis buffer (50 mM sodium phosphate buffer, pH 7.5, 5% w/v glycerol, 300 mM KCl, 1 mM MgCl₂, 0.2% w/v Tween-20, 10 mM BME, 0.1 mM ATP, 1 × protease inhibitor cocktail, 30 mM Imidazol) was bound to Ni-NTA resin. Proteins were eluted from the column by proteolytic cleavage using PreScission Protease cleaving at a 3C site located between the GFP and the His-tag. Kinesin-5 (human Kif11, for data from Supplementary Figure 3) was expressed using the flashBAC System (Oxford Expression Technologies). Protein lysate in lysis buffer was bound to Ni-NTA resin and eluted with 300 mM imidazole in lysis buffer. Protein aliquots were flash-frozen and stored at − 80 °C. Recombinant histidine-tagged full-length *Schizosaccharomyces pombe* Ase1-GFP and full-length *Drosophila melanogaster* GFP-Ncd (for data from Fig. 2g) were expressed and purified previously[19].

**Silica microsphere functionalization**. The 1 μm diameter, carboxylated silica microspheres (Bangs Laboratories, #SC04N) were functionalized with NeutrAvidin (Thermo Scientific) using EDC-NHS chemistry (1-ethyl-3-(-3-dimethylamino-propyl) carbodiimide hydrochloride (EDC) and *N*-hydroxysuccinimide (NHS)). In detail, 5 mg dry silica microspheres were dispersed in 900 μl of 100 mM sodium

carbonate buffer, pH 9.6 and sonicated for 10–30 min until single microspheres were visible under the microscope. Microspheres were then washed by subsequent steps of centrifuging (1 min, ~400 × g) and resuspending in 900 μl 2-(N-morpholino)ethanesulfonic acid (MES) buffer, pH 6.5, that was supplemented with 0.05% Tween-20 (Merck). Final resuspension volume was 150 μl MES without Tween-20. This was supplemented with 150 μl MES buffer supplemented with 1.6% EDC and 0.8% NHS (w/v) and incubated for 20 min at 37 °C in a shaker to prevent microsphere aggregation. Microspheres were then washed 4 times with MES buffer supplemented with 0.1% Tween-20 and resuspended in 150 μl 50 mM borate buffer, pH 8.55. 100 μl 1 mg ml$^{-1}$ NeutrAvidin in borate buffer was added and incubated for 20 min at 37 °C, 1–2 h at room temperature and at least overnight and up to several days at 4 °C. The reaction was quenched by adding 30 μl 0.8 M Tris buffer at pH 7.4 and incubating at room temperature for 90 min. Microspheres were washed in 10 mM sodium phosphate buffer, pH 7.4 supplemented with 0.1 mg ml$^{-1}$ bovine serum albumin (BSA) and 0.05% Tween-20. Microspheres were either used directly or stored in 10 mM sodium buffer supplemented with 0.1 mg ml$^{-1}$ BSA and 0.05% Tween-20, 5% glycerol and 0.1% sodium azide. If stored, the microspheres were washed at least 5 times before use.

**Motility assays**. Assay buffer consisted of 20 mM Hepes at pH 7.2, 1 mM EGTA, 2 mM MgCl₂, 75 mM KCl, 10 μM paclitaxel, 0.5 mg ml$^{-1}$ casein, 10 mM dithiothreitol, 0.1% Tween-20, 20 mM D-glucose, 22.4 μg ml$^{-1}$ glucose oxidase, 20 μg ml$^{-1}$ catalase and either 1 mM ATP or 1 mM adenosine diphosphate (ADP).

For assays using surface-immobilized GFP-Snap-Ncd motors (NcdΔtail), 2.5–30 μg ml$^{-1}$ anti-SNAP antibodies (Antikoerper-online, Aachen, Germany, ABIN1573927) in phosphate-buffered saline (PBS) were allowed to bind to diphenyldimethoxysilane (DDS)-functionalized coverslips. The surfaces were then passivated using pluronic F127, 1% w/v in PBS. NcdΔtail motors were injected at saturating concentration (12.8 nM) in the flow chamber and allowed to bind to the antibodies, washed out and microtubules were added. Following passivation, all steps were performed in ATP assay buffer.

For experiments using crosslinked microtubules (without Ase1), microtubules, silica microspheres and flow chambers were prepared as described previously[20]: anti-digoxigenin antibodies (Roche, 11333089001, 20 μg ml$^{-1}$ in PBS) were allowed to bind to DDS-functionalized coverslips. The surfaces were consecutively blocked with 1% Pluronic F127 (Sigma, P2443) in PBS for at least 1 h. Afterwards, digoxigenated, dimly cy5-labeled microtubules were injected into the flow chamber and bound to the surface-immobilized antibodies. Then, 0.4 nM flNcd in ADP assay buffer was injected into the flowcell and allowed to bind to the microtubules. Consecutively, biotinylated, brightly cy5-labeled microtubules were injected and allowed to form microtubule pairs in ADP assay buffer. Unbound microtubules were washed out using ADP assay buffer. The buffer was then replaced with ATP assay buffer. When optical tweezers were used, the ATP buffer was supplemented with NeutrAvidin-functionalized silica microspheres. Some of the microtubule velocity measurements were conducted using a different set of microtubules, namely dimly rhodamine-labeled, weakly biotinylated microtubules for surface immobilization via anti-biotin antibodies and brightly rhodamine-labeled microtubules as transported microtubules. Kinesin-5 (Kif11) sliding assays were conducted in exactly the same manner, including the use of the same buffers as flNcd sliding experiments. Experiments using Ase1 have been conducted previously[20].

For the microtubule buckling assay, the surface was modified with anti-digoxigenin antibodies and two kinds of microtubules were allowed to bind: dimly cy5-labeled, digoxigeninated as well as rhodamine-labeled, biotinylated, digoxigeninated microtubules. After washing out unbound microtubules, NeutrAvidin was introduced and allowed to bind to the subset of biotinylated microtubules. Subsequently 0.4 nM flNcd followed by transported microtubules (brightly cy5-labeled, biotinylated) were introduced in ADP assay buffer. Then, ATP assay buffer was added. Transported microtubules bound to cy5-labeled surface-immobilized microtubules would start sliding until encountering a NeutrAvidin molecule bound to a biotinylated surface-immobilized microtubule.

All experiments were performed over several months, with data replicates acquired during multiple independent experimental days.

**Image acquisition**. Microscopy data for gliding velocity evaluation and part of the data for sliding velocity evaluation were acquired using an Axio Observer Z3 microscope (Carl Zeiss AG, Germany) equipped with a 63×, 1.46 NA, WD = 0.1 mm objective (plan, apochromat, Zeiss) in combination with an EMCCD iXon-3 camera (Andor, Northern Ireland) and MetaMorph software (Molecular Devices, USA). Samples were excited using an Omicron (Rodgau, Germany) laser of 488 nm and a Prior Lumen200 arc lamp. The objective temperature was controlled to be 29 °C, thereby stabilizing the sample temperature. Cy5-labeled microtubules and GFP-labeled proteins were visualized sequentially by switching between two laser lines and switching filter sets (Chroma Technology). Acquisition rates were one frame per 1–2 s.

Optical tweezers experiments were conducted using a NanoTracker (JPK, Berlin), which was based on a Nikon Eclipse Ti equipped with a TIRF module (Nikon). We used a 60×, 1.49 NA, WD = 0.12 mm oil objective (CFI Apo, Nikon) for trapping and visualization in combination with an EMCCD iXon-3 camera (Andor, Northern Ireland) and VisiView software (Visitron, Germany). Samples

were excited using diode lasers of 488 nm and 642 nm wavelength (Stradus Vortran Sacramento, USA). Cy5-labeled microtubules and GFP-labeled proteins were visualized sequentially by switching between two laser lines and using a two-band filter set (Chroma technology). Acquisition rates were one frame per 1–2 s. Objective temperature was not controlled actively at the optical tweezers setup. However, sliding velocities from the temperature controlled setup and the optical tweezers setup were not significantly different (Kolmogorov–Smirnov test: $p = 0.83$).

**Image analysis**. Ncd motor domain surface density for the assays using surface-immobilized Ncd was calculated from fluorescence intensity. To that end, the surface fluorescence intensity density was divided by the fluorescence intensity of a single GFP-labeled kinesin-1. The motor surface density was established for all the antibody concentrations used prior to the experiments. Microscopy data from optical tweezers data were used to quantify the number of fluorescently labeled flNcd molecules in the microtubule overlap in situ. The fluorescent intensity was quantified using ImageJ, manually selecting areas with single molecules from an assay with a 10 times lower concentration of flNcd added on single microtubules and subtracting background. Microscopy data were also used to assure that only two, not multiple, microtubules were crosslinked.

FIESTA[36] tracking was used for microtubules propelled by surface-immobilized Ncd motor domains, yielding velocity and microtubule length. FIESTA kymograph evaluation was used for crosslinked microtubules, yielding velocities. The microtubule length was quantified by measuring the length by hand in FIESTA. In both cases the data were time weighted: therefore, tracked data which yielded instantaneous velocities were averaged prior to time weighting to allow for an easier comparison between the two data sets as velocities stemming from kymograph evaluation are inherently averaged.

In the case of geometrically hindered diffusion, the microtubules swiveled a lot, which made kymograph evaluation impossible. Therefore, the distance of either the trailing or leading end of the microtubule to the pivoting point (usually a microtubule crossing) was marked by hand in FIJI[37,38] and the traveled distance in between the frames was then divided by the time difference between frames to calculate instantaneous velocities. Velocities were averaged for each microtubule and then time weighted to allow for better comparison with the other data sets.

**Optical tweezers experiments**. The optical trap setup (JPK, Berlin) was used with a 60×, 1.49 NA, WD = 0.12 mm oil objective (CFI Apo, Nikon) for trapping and visualization and a 60×, 1.2 NA, WD = 0.28–0.31 mm water objective (CFI Plan APO, Nikon) for back-focal plane detection of the forward scattered laser light which was projected on a quadrant photo diode. Distance and force were calibrated using the built-in power spectrum routine from the JPK software. Laser powers between 0.5 and 1 W were used, corresponding to trap stiffness between 0.05 pN nm$^{-1}$ and 0.1 pN nm$^{-1}$.

A functionalized NeutrAvidin-coated silica microsphere was caught with the optical tweezers from solution. It was visually ensured that it was a single microsphere. Every microsphere was calibrated using the built-in routine of the JPK software at a distance from the surface of around 3–5 μm. For the force measurements on microspheres propelled by surface-immobilized Ncd motor domains, the microsphere was lowered and brought into the proximity of a moving microtubule. Microspheres were carefully moved close to a microtubule, but not directly over it, as there was some crosstalk between $x$–$y$ displacement and $z$-position signal. The microsphere was first brought to an appropriate height and then moved laterally for the remaining few micrometers to bind to the moving microtubule. This ensured that the force prior to microsphere attachment could be used to offset the overall force. Once the microsphere was in close proximity, it often bound very quickly to the moving microtubule. This was confirmed by a fast increase in measured force (see Fig. 1c) and often halting of the gliding microtubule.

For the force measurements on microtubules propelled by diffusively anchored flNcd motors, crosslinked to surface-immobilized microtubules, the procedure was very similar. However, when the microsphere was lowered, it was lowered directly on top of the moving microtubule to prevent pushing the transported microtubule off the surface-immobilized microtubule. Once the microsphere was in close proximity to the transported microtubule, it often bound to the transported microtubule which was confirmed by one or several of the following observations/methods. (i) The transported microtubule stopped. (ii) The optical tweezers laser was turned off and the microsphere was monitored in bright field. When bound to a microtubule, it did display minimal wiggling, but no free diffusion. (iii) The stage was moved along the long axis of the surface-immobilized microtubule with the tweezers engaged. When the microsphere was bound to the transported microtubule, the microtubule stayed in the same spot, moving relative to the surface-immobilized template microtubule. Force-velocity measurements were conducted by either setting a constant velocity and monitoring the force or, more often, setting a constant force and monitoring the velocity needed to maintain that force. Force and velocity were always applied in forward and backward direction and no difference was visible in magnitude.

The flNcd-driven transport was not degraded by the microsphere attachment or the repetitive force-velocity scans, as microtubule sliding velocities before and after microsphere attachment and conducting force-velocity scans did not differ significantly ($N = 26$ microtubules, Kolmogorov–Smirnov test: $p = 0.14$, Supplementary Figure 4).

Force measurements on non-aligned microtubules that were transported by flNcds and microtubules that were transported by microtubule-bound NcdΔtail were conducted as for those microtubules transported by surface-immobilized NcdΔtail.

All experiments were performed over several months. Data for all experiments were acquired during at least five independent experimental days.

**Force evaluation**. Force data were evaluated using custom-written Matlab (MathWork, USA) routines. When forces were exerted by surface-immobilized Ncd motor domains and by flNcd motors between non-aligned microtubules, the binding of the microsphere and subsequent force exertion were clearly visible. For these two cases, a time period of several seconds prior to microsphere attachment was used to determine the force offset in $x$- and $y$-direction by averaging. Averaging over the force values when the microsphere displacement had reached a plateau determined the stopping force in $x$- and $y$-direction. Pythagoras' theorem was used to calculate the absolute stopping force in an arbitrary direction. The graphs in Figs. 1c, 3d, g display absolute force. Values fluctuating below zero are thus mirrored to be above zero. However, once the stopping force is reached, the fluctuations were far away from zero force, thus not affecting the value of the averaged stopping-force.

Forces between aligned crosslinked microtubules were often too small to distinguish a binding event by eye. We conducted force-velocity measurements either using the force-feedback routine of the JPK software or a constant velocity routine. Both methods gave comparable results. Each data trace was carefully checked by eye for signs of unspecific surface interaction and were rated for their quality (how constant the velocity was) as a weight for the later fit. All data collected from one microtubule pair were used to create a linear force-velocity fit. This fit was used to determine the stopping force as shown in Supplementary Figure 3. In short, the stopping force is the difference in force between the force values that correspond to no-load transport velocity and no movement (stop) in the fit. The no-load transport velocity was measured independently either before or after the force-velocity measurements and there was no significant difference between velocities measured before or after microsphere attachment (Supplementary Figure 4).

Forces quantified using the entropic force of Ase1 were calculated using the one-dimensional ideal gas equation $F = N \times k_B \times T / L$. This approximation holds true for a dilute gas, which was always the case.

The buckling force was calculated using the following formula: $4\pi \times EI /(L^2)$ with EI being the flexural rigidity with the value[21] $EI = 6 \times 10^{-24}$ nm$^2$ and $L = 8.3$ μm and $L = 10.0$ μm.

**Data presentation**. In boxplots (Figs. 1c, 1f, 2c and 4, Supplementary Figures 3 and 4) red midline indicates the median; bottom and top box edges indicate the 25th and 75th percentiles, respectively; the whiskers extend to the most extreme data points not considered as outliers; the outliers (red crosses) are defined as data points greater than q3 + 1.5(q3 – q1) or less than q1 – 1.5(q3 – q1), where q1 and q3 are the 25th and 75th percentiles, respectively.

**Data availability**. All relevant data are available from the authors upon reasonable request.

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

## Acknowledgements

We thank Corina Bräuer, Eva Zatecka and Aniruddha Mitra for technical assistance, Regis Lemaitre and Aliona Bogdanova at the MPI-CBG Protein Expression Facility for providing the DefBac baculovirus expression system and compatible expression vectors, respectively, Hauke Drechsler for comments on the manuscript and Michael Thompson for language editing. A.L., A.-M.S., M.B. and S.D. acknowledge financial support from the Dresden International Graduate School for Biomedicine and Bioengineering (stipend to A.L.) and the German Excellence Initiative (TU Dresden Support-the-Best grant to S.D.). Z.L. and M.B. acknowledge financial support from the Czech Science Foundation (grant numbers 18-08304s and 17-12496Y), the project BIOCEV (CZ.1.05/1.1.00/02.0109) from the ERDF and CAS (RVO: 86652036).

## Author contributions

A.L., M.B., Z.L. and S.D. designed the experiments; A.L., A.-M.S, Z.L. and M.B. performed the experiments; A.L. and A.-M.S. analyzed the data; A.L. M.B., Z.L and S.D. wrote the manuscript.

## Additional information

**Competing interests:** The authors declare no competing interests.

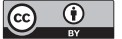

