## [Peer Review File · Nature Communications]

Reviewers' comments:

Reviewer #1 (Remarks to the Author):

The authors investigate the ability of ncd motors to cooperate to generate high forces as a population and they characterize the role the microtubule-binding tail, which diffuses along the microtubule, plays in the ability of ncd motors to exert forces when sliding microtubules relative to one another. The measurement of different velocities when ncd is bound tightly to surface versus bound through tail to an immobilized microtubule is a clear and interesting result. And the measurement of summed forces for surface-bound motors and much weaker forces for motors bound to immobilized microtubules through their tails is also interesting and clear. The clean message is that because the tail binds quite diffusively, rather than more rigidly, which has been the general model in the field to date, ncd is actually a weak transporter and microtubule slider, and is more of a microtubule bundler.

Overall the paper was well written and the experiments carefully conducted. I had some complaints about the data presentations, noted below. It seemed like this was perhaps written as a NCB article that was then cascaded to Nat Comm, and so the text and figs don't need to be in the austere space-limited NCB form and should be expanded.

1) For such quantitative data, I felt the presentation in Fig 1B was weak with the arbitrary bin sizes and spacings. I much prefer fig S1A, with the change that the bins all be the same 1 um width and spaced equally, with a single bin >10 um or whatever you like. Or, make bin 3 um wide and do it that way. As is in 1B, it looks like selective selection of data, but the reality is that the data are clear and the line is flat for all mt lengths. I would extend this suggestion to Fig 2B, which is similar and the idea to Fig 1D, where the motor number bins similarly seem arbitrary and subjective.

2) My chief complaint is Figure 2. The data in Fig S3, which is the important raw data, and S4, which is necessary to understand the F_s interpolation approach used. should be put in the main text here. As is, the text almost entirely relates to the supplementary and there are only 3 figures anyway in the whole main text. Probably the Ase1 and bucklink should then be its own new figure. Please be more thorough with your data presentation.

3) The result in figure 3 is outstanding. I think that the presentation could be improved by a better setup in the text. The result from fig 1 and 2 tell the story about immobilized vs diffusive (I would suggest summing up this story after presentation of figure 2). This makes predictions for fig 3 data, and a good way to really sell the fig 3 would be to lay out these predictions and then hammer home the results. A minor complaint – I don't like the terms "diffusing", on x-axis in fig 3C and D, they don't stick in the readers' brain as a description (I do like the pictures). Maybe "Diffusively-bound, parallel" and "diffusively-bound perpendicular" or something like that would be better terms.

4) I suggest a final model figure that goes with the discussion and gives the reader a clear picture of what you're talking about and what ncd can and can't do in the spindle.

Minor comments

Instead of calling them "motor domains", which is very general, I suggest dimeric or tail-less dimer or something more specific that clarifies that they are dimeric and have no tail (motor domain implies only monomeric head).

Fig S3. This figure shows an interesting effect, but it is hard to see. I suggest stacking x, y, and z

vertically to make time axis the same, and just indicate with an arrow where the stage shift occurred (the amplitude of stage doesn't mean much and the x and z stage change is nothing).

Fig S4 legend. Please be more scientific about "good" data. There should be a statistical rule that is used and a different wording. This figure is important but difficult to understand, I suggest clarifying the entire legend.

P5 - "finite angles" should be "defined angles" as there is no such thing as an infinite angle.

Reviewer #2 (Remarks to the Author):

This is interesting and carefully done. It reveals new aspects of the kinesin-14 mechanism. I have only minor comments.

1. The tendency for the kinesin-14 tail to diffuse on MTs is well established. Nonetheless it would be good to see mass spec of the expressed and protease-treated his tag full length Ncd protein, to confirm all is as it should be. The experiments were done at fairly low ionic strength (20 mM HEPES, 70mM KCl). During optimisation, was it seen that the tendency for the tail domain of Ncd-FL to diffuse can be adjusted by varying the ionic strength?

2. The MT sliding velocities measured relate to an estimated number of engaged motors per MT. It would be good to estimate the density of engaged motors. MT sliding velocity is insensitive to MT length (Fig 2b) showing that under these conditions the number of engaged motors makes no difference to velocity, but it is unclear whether the motor surfaces are relatively sparse, or very dense. Please comment. It would be good to quote the surface density, since it was measured for the various antibody surfaces, and an estimate of the range of spacings between motors on the surfaces, to see how this relates to the estimates for the numbers of engaged motors and the densities of engaged motors. Earlier work suggested that for some kinesin-14 motors at least, MT sliding assays slow down as the motor surface gets crowded.

3. "stall force" is used for both the force required to stop MT gliding when the Ncd tails are fixed to the coverlips, and the force required to stop MT sliding when the motors are crosslinking and their anchor domains slip. To my mind in this second situation the motors don't get the chance to stall and it is confusing to call this a stall force. A better term is needed to describe the force that tail domain binding can support without slipping - perhaps "anchor domain slippage force" or something like that.

4. Top of p4 "we conducted force-velocity measurements" please briefly outline how, in the main text. I think what was done was to gingerly attach a trapped bead to a sliding MT and then record subsequent MT sliding with the trap parked, so that the hindering load gradually increased as the MT slid and the MT gradually slowed down. However high laser powers / trap stiffnesses were used and the example shown it looks like MTs slid at constant rate (perhaps the unloaded rate?) and then suddenly stopped - is that right? And typical? Please clarify how much displacement of the MT corresponded to the forces graphed in Fig 1C.

5. Conclusions. As the authors discuss, the Ncd tail domain binds EB1 and tip tracks at MT plus ends. The authors do not discuss the possibility that the interesting property they have discovered (that full length ncd crosslinks and slides orthogonally-arranged MTs quickly and with appreciable force, but slips and so is unable to do useful work when the MTs are in line) might be relevant in vivo. Perhaps this might make Ncd especially good at pole focussing?

Reviewer #3 (Remarks to the Author):

In the current paper the authors characterize the motility and force characteristics of the Kinesin-14 Ncd. They show that a truncated version of Ncd, which is bound to a glass surface via an antibody/tag linkage results in gliding velocities of 270 nm/s and forces of several piconewtons that scaled with the number of motors. In contrast, using full-length Ncd in between microtubule cross-links resulted in a four-fold reduction of motor velocity and a dramatic decrease in force to less than one piconewton. Analyzing the velocity of movement of microtubules that were cross-linked at a fixed angle resulted in intermediate velocities and increased stall forces. From this data, they conclude that the diffusive tail binding of Ncd in microtubule cross-links renders the motor ineffective at producing significant force.

The authors present some intriguing observations, which if correct, will make people in the field think about how Kinesin-14 proteins and other motors act in the spindle. However, the current data are quite preliminary, and other models could explain the data. In addition, the authors present observational data and do not move forward to try and test the models that they propose. The paper is therefore unsuitable for publication at this time without significant additional experimentation. Below are a few of the major concerns for the authors to think about as they move the project forward.

1. It was previously shown that ncd will cross-link and slide anti-parallel microtubules and make static cross-links on parallel microtubules. In the present set-up, the authors make the assumption that all of the sliding is due to anti-parallel microtubule sliding, which has not been confirmed in this study. In addition, the authors don't take into account the geometry of the motors in the cross-links and whether any of the motors could produce a drag-force depending on orientation and engagement.
2. The authors present a model that proposes that the change in force and motility is due to the diffusive motions of the tail. This is by inference by comparing truncated motors bound to glass via an antibody linkage compared to full-length motors cross-linking microtubules. To test this model the authors need to come up with a way or mutating the tail to lock it onto the microtubule and then show that it really is the diffusive behavior of the tail that is responsible for the change in force in a cross-linking assay.
3. The experiments with the non-aligned microtubules could increase the velocity due to reduction in diffusion or to limitation in non-productive interactions as described in point 1 above. Therefore, these experiments are correlative at the present time.
4. On a more positive note- I love the analogy in the discussion to pushing a car while on roller skates- this is fabulous!!!!

We thank the reviewers for their valuable recommendations. Addressing all of the comments, we altered the manuscript accordingly.

Please find our detailed replies (in blue) to the reviewers' comments (in black) and the actions taken (in bold blue) below. In the revised manuscript and supplementary information the new text and figures are marked in blue.

Reviewer #1 (Remarks to the Author):

The authors investigate the ability of ncd motors to cooperate to generate high forces as a population and they characterize the role the microtubule-binding tail, which diffuses along the microtubule, plays in the ability of ncd motors to exert forces when sliding microtubules relative to one another. The measurement of different velocities when ncd is bound tightly to surface versus bound through tail to an immobilized microtubule is a clear and interesting result. And the measurement of summed forces for surface-bound motors and much weaker forces for motors bound to immobilized microtubules through their tails is also interesting and clear. The clean message is that because the tail binds quite diffusively, rather than more rigidly, which has been the general model in the field to date, ncd is actually a weak transporter and microtubule slider, and is more of a microtubule bundler.

Overall the paper was well written and the experiments carefully conducted. I had some complaints about the data presentations, noted below. It seemed like this was perhaps written as a NCB article that was then cascaded to Nat Comm, and so the text and figs don't need to be in the austere space-limited NCB form and should be expanded.

Authors: We thank the reviewer for the positive remarks. We improved the data presentation according to the suggestions (see also below).

1) For such quantitative data, I felt the presentation in Fig 1B was weak with the arbitrary bin sizes and spacings. I much prefer fig S1A, with the change that the bins all be the same 1 μm width and spaced equally, with a single bin $>10 \mu\text{m}$ or whatever you like. Or, make bin 3 μm wide and do it that way. As is in 1B, it looks like selective selection of data, but the reality is that the data are clear and the line is flat for all mt lengths. I would extend this suggestion to Fig 2B, which is similar and the idea to Fig 1D, where the motor number bins similarly seem arbitrary and subjective.

Authors: We re-plotted the data according to the reviewer's suggestion.

2) My chief complaint is Figure 2. The data in Fig S3, which is the important raw data, and S4, which is necessary to understand the F_s interpolation approach used. should be put in the main text here. As is, the text almost entirely relates to the supplementary and there are only 3 figures anyway in the whole main text. Probably the Ase1 and bucklink should then be its own new figure. Please be more thorough with your data presentation.

Authors: We now show Figures S3 and S4 as a part of the main Figure 2. However, we kept the Ase1 data in the Figure 2 as we believe it will be easier for the reader to compare the forces in the panels 2f and 2g if they are part of the same figure.

3) The result in figure 3 is outstanding. I think that the presentation could be improved by a better setup in the text. The result from fig 1 and 2 tell the story about immobilized vs diffusive (I would suggest summing up this story after presentation of figure 2). This

makes predictions for fig 3 data, and a good way to really sell the fig 3 would be to lay out these predictions and then hammer home the results. A minor complaint – I don't like the terms “diffusing”, on x-axis in fig 3C and D, they don't stick in the readers' brain as a description (I do like the pictures). Maybe “Diffusively-bound, parallel” and “diffusively-bound perpendicular” or something like that would be better terms.

Authors: We have rephrased the text according to the reviewer's suggestion and we moved the comparison of velocities and forces for different scenarios to the new Figure 4. Since we performed a new set of experiments with motors fixed to a microtubule, we now have four distinct scenarios in this figure. We have thus changed the figure descriptions to "diffusible anchoring on microtubule", "diffusible anchoring on microtubule - geometrically restricted", "static anchoring on microtubule" and "static anchoring on surface".

4) I suggest a final model figure that goes with the discussion and gives the reader a clear picture of what you're talking about and what ncd can and can't do in the spindle.

Authors: We now added a figure summing up the findings of our paper (new Figure 5) as suggested by the reviewer.

Minor comments

Instead of calling them “motor domains”, which is very general, I suggest dimeric or tail-less dimer or something more specific that clarifies that they are dimeric and have no tail (motor domain implies only monomeric head).

Authors: We now refer to the truncated, tail-less, SNAP_F-tagged, dimeric Ncd-motor constructs as Ncd Δ tail.

Fig S3. This figure shows an interesting effect, but it is hard to see. I suggest stacking x, y, and z vertically to make time axis the same, and just indicate with an arrow where the stage shift occurred (the amplitude of stage doesn't mean much and the x and z stage change is nothing).

Authors: We have replotted Figure S3 (which is now part of the new Figure 2) according to the reviewer's suggestion.

Fig S4 legend. Please be more scientific about “good” data. There should be a statistical rule that is used and a different wording. This figure is important but difficult to understand, I suggest clarifying the entire legend.

Authors: We included Figure S4 into Figure 2 and clarified the procedure in the text. The data was rated visually by how constant the stage velocity was, more specifically, it was judged how well the stage position data was in agreement with a linear fit. We did not use a statistical rule, such as a minimum R-value, because a short but high deviations (surface sticking) with the right slope would yield an R-value of similar magnitude as a long but small deviation with the wrong slope. Developing a novel procedure at $N=33$ seemed similarly arbitrary as judging by plain eyesight.

P5 – “finite angles” should be “defined angles” as there is no such thing as an infinite angle.

Authors: We have rephrased the text and we now refer to these microtubules as "non-aligned" microtubules.

Reviewer #2 (Remarks to the Author):

This is interesting and careful done. It reveals new aspects of the kinesin-14 mechanism.

Authors: We thank the reviewer for this positive assessment.

I have only minor comments.

1. The tendency for the kinesin-14 tail to diffuse on MTs is well established. Nonetheless it would be good to see mass spec of the expressed and protease-treated his tag full length Ncd protein, to confirm all is as it should be.

Authors: We are aware of the fact that the presence of the his-tag can affect the binding of microtubule-associated proteins to microtubules. To circumvent this pitfall and to ensure that the his-tag is indeed cleaved off in our protein preps we purified the Ncd constructs as follows: Upon binding of the his-tagged protein to the Ni-NTA resin, we did not elute the protein. Rather, we used the PreScission protease to cleave the motor protein from the his-tag, which was still bound to the Ni-NTA resin (cleaving directly in the column). We then collected the cleaved off protein by a wash step without any elution reagent. By this approach we make sure that all collected protein is his-tag less.

We now clearly describe this procedure in the Methods.

The experiments were done at fairly low ionic strength (20 mM HEPES, 70mM KCl). During optimisation, was it seen that the tendency for the tail domain of Ncd-FL to diffuse can be adjusted by varying the ionic strength?

Authors: We performed initial experiments at different salt concentrations and we noticed that the interaction of the tail domain with microtubules is indeed dependent on the ionic strength, similarly to the human kinesin-14 (Braun et al, Nat Chem Biol 2017, 13(12), 1245-1252). The buffer composition that we use throughout the paper was optimized for the formation of microtubule bundles.

2. The MT sliding velocities measured relate to an estimated number of engaged motors per MT. It would be good to estimate the density of engaged motors. MT sliding velocity is insensitive to MT length (Fig 2b) showing that under these conditions the number of engaged motors makes no difference to velocity, but it is unclear whether the motor surfaces are relatively sparse, or very dense. Please comment.

Authors: Please see our answer below.

It would be good to quote the surface density, since it was measured for the various antibody surfaces, and an estimate of the range of spacings between motors on the surfaces, to see how this relates to the estimates for the numbers of engaged motors and the densities of engaged motors. Earlier work suggested that for some kinesin-14 motors at least, MT sliding assays slow down as the motor surface gets crowded.

Authors: We estimated the density of the motors from the integrated intensity of fluorescence along the length of the microtubules both, in our sliding and gliding experiments. Both results show a similar, rather low, density range of about 1 motor per 100 nm of the microtubule length. For the gliding assays the number of motors that are actually bound to the microtubule is hard to estimate, however the fact that we often see short (approximately less than 0.5 micrometer) microtubules detaching from the surface during gliding, suggests that the density of engaged motors is indeed in the low range estimated by the fluorescence intensity. At these densities it is rather unlikely that

motors come in contact with each other, which could decrease their velocity. **We now comment on this in the discussion section.**

3. "stall force" is used for both the force required to stop MT gliding when the Ncd tails are fixed to the coverlips, and the force required to stop MT sliding when the motors are crosslinking and their anchor domains slip. To my mind in this second situation the motors don't get the chance to stall and it is confusing to call this a stall force. A better term is needed to describe the force that tail domain binding can support without slipping - perhaps "anchor domain slippage force" or something like that.

Authors: We agree that "stall force" is a misleading term in this context. **We now refer to the force at which the microtubule stopped in both experiments (gliding and sliding) as the "stopping force". Moreover, in the discussion we now comment on the fact that in the situation, when the tail domains can slip, this force is related to the frictional force of the tail domains rather than the maximum force of the motor domain.**

4. Top of p4 "we conducted force-velocity measurements" please briefly outline how, in the main text. I think what was done was to gingerly attach a trapped bead to a sliding MT and then record subsequent MT sliding with the trap parked, so that the hindering load gradually increased as the MT slid and the MT gradually slowed down. However high laser powers / trap stiffnesses were used and the example shown it looks like MTs slid at constant rate (perhaps the unloaded rate?) and then suddenly stopped - is that right? And typical? Please clarify how much displacement of the MT corresponded to the forces graphed in Fig 1C.

Authors: We have now added a section explaining the force velocity measurements into the main text. As microtubules stopped immediately upon bead attachment (see new Figure 2d), we used the force-feedback routine provided by the instrument's manufacturer to quantify the relationship between force and velocity for microtubule pairs. To generate signals above noise level, we had to go to velocities well over microtubule sliding velocities of around 200–1000 nm/s. We have, however, not exceeded the force range in which other kinesin motors operate.

5. Conclusions. As the authors discuss, the Ncd tail domain binds EB1 and tip tracks at MT plus ends. The authors do not discuss the possibility that the interesting property they have discovered (that full length ncd crosslinks and slides orthogonally-arranged MTs quickly and with appreciably force, but slips and so is unable to do useful work when the MTs are in line) might be relevant in vivo. Perhaps this might make Ncd especially good at pole focussing?

Authors: We thank the reviewer for this interesting idea. **We took the liberty to mention this possibility now in the discussion.**

Reviewer #3 (Remarks to the Author):

In the current paper the authors characterize the motility and force characteristics of the Kinesin-14 Ncd. They show that a truncated version of Ncd, which is bound to a glass surface via an antibody/tag linkage results in gliding velocities of 270 nm/s and forces of several piconewtons that scaled with the number of motors. In contrast, using full-length Ncd in between microtubule cross-links resulted in a four-fold reduction of motor velocity and a dramatic decrease in force to less than one piconewton. Analyzing the

velocity of movement of microtubules that were cross-linked at a fixed angle resulted in intermediate velocities and increased stall forces. From this data, they conclude that the diffusive tail binding of Ncd in microtubule cross-links renders the motor ineffective at producing significant force.

The authors present some intriguing observations, which if correct, will make people in the field think about how Kinesin-14 proteins and other motors act in the spindle. However, the current data are quite preliminary, and other models could explain the data. In addition, the authors present observational data and do not move forward to try and test the models that they propose. The paper is therefore unsuitable for publication at this time without significant additional experimentation. Below are a few of the major concerns for the authors to think about as they move the project forward.

Authors: We thank the reviewer for the positive remark about the general interest of our observations. Following the valuable suggestions of the reviewer, **we performed additional experiments and significantly extended the discussion of our results (see below for details).**

1. It was previously shown that ncd will cross-link and slide anti-parallel microtubules and make static cross-links on parallel microtubules. In the present set-up, the authors make the assumption that all of the sliding is due to anti-parallel microtubule sliding, which has not been confirmed in this study.

Authors: As the reviewer notes, the fact that kinesin-14 crosslinks both antiparallel and parallel microtubules, but slides only antiparallel ones is well established (Fink et al, Nat Cell Biol. 2009 Jun;11(6), 717-23; Braun et al, Nat Cell Biol. 2009 Jun;11(6):724-30; Braun et al, Nat Cell Biol. 2011 Sep 4;13(10), 1259-64; Hentrich & Surrey, J Cell Biol. 2010 May 3;189(3):465-80). We have shown this effect previously (Fink et al, Nat Cell Biol. 2009 Jun;11(6), 717-23) using the very same protein construct and the same buffer conditions that we use in the current paper. While we therefore do not find it appropriate to include this data again in the current paper **we do refer to this behavior now clearly in the main text when describing the sliding assays.**

In addition, the authors don't take into account the geometry of the motors in the cross-links and whether any of the motors could produce a drag-force depending on orientation and engagement.

Authors: We thank the reviewer for bringing up this comment. Indeed, we performed the sliding assays with full-length motors, whereas the gliding assays were performed with truncated constructs lacking the tail domain. Also, the motors in the sliding assay can move within the overlap, whereas in the gliding assay the motors are firmly attached to the coverslip surface. The motors in the sliding experiments thus indeed might produce a drag-force, which is not present in the gliding geometry stemming either from the tail-domain, which in the sliding assay might get to a close proximity of the motor domain, or from the "collisions" between the motors in the sliding overlap.

Among other considerations, we performed additional analysis of our sliding experiments and now show that the density of motors in the microtubule overlaps is rather low (~1 motor per 100 nm length), which makes collisions between the motors rather unlikely. We also never observed the formation of motor clusters in the overlaps, suggesting that the motors did not hinder each other at the concentrations used in our experiments. Inhibition of the motor domain by the tail domain in the sliding assays (being another potential caveat) is ruled out as the cause of the slower velocities in the

sliding experiments because non-aligned microtubules slid at high speeds (in the same experiment, internal control). In summary, our experiments do not support the hypothesis of an additional drag-force (or other related effects) in the sliding geometry. **We now discuss these new findings in detail in a new section at the beginning of the Discussion.**

2. The authors present a model that proposes that the change in force and motility is due to the diffusive motions of the tail. This is by inference by comparing truncated motors bound to glass via an antibody linkage compared to full-length motors cross-linking microtubules. To test this model the authors need to come up with a way or mutating the tail to lock it onto the microtubule and then show that it really is the diffusive behavior of the tail that is responsible for the change in force in a cross-linking assay.

Authors: We thank the reviewer for this suggestion. To directly show that it is the diffusive behavior of the tail which leads to diminished velocities in the sliding assays, **we have now performed a new set of experiments, in which we lock the kinesin-14 motors to the microtubule lattice (using snap-tag-motor bound to BG-labeled microtubules) and investigated the sliding of two aligned microtubules relative to each other by such firmly-attached motors (part of the new Figure 3).** In these new experiments microtubules slid relative to each other at velocities, which are significantly higher than the velocities driven by the native motors (with the slippery tails). These experiments thus further strengthen our hypothesis that it is the diffusivity of the tail domain that sets the maximum force that can be generated by ensembles of kinesin-14 motors.

3. The experiments with the non-aligned microtubules could increase the velocity due to reduction in diffusion or to limitation in non-productive interactions as described in point 1 above. Therefore, these experiments are correlative at the present time.

Authors: As described above we have analyzed the densities of the motors in the overlaps, which suggests that the motors rarely come into contact. We do not see any buildup of the GFP signal in the crosses between non-aligned microtubules. The motor density in these regions is in the same range as in the aligned overlaps. Our analysis thus suggests that non-productive interactions stemming from the motors coming to contact with each other are rather unlikely also in the crosses between non-aligned microtubules. **We have now added a new paragraph in the Discussion section, covering this topic.**

4. On a more positive note- I love the analogy in the discussion to pushing a car while on roller skates- this is fabulous!!!!

Authors: We thank the reviewer for this positive comment. **Triggered by the remark we now also represent this analogy in the new Figure 5.**

REVIEWERS' COMMENTS:

Reviewer #1 (Remarks to the Author):

The authors have responded fully to my comments and I commend their new experiments.

Reviewer #2 (Remarks to the Author):

I have carefully read the revised ms and the author's responses to my comments and can confirm that they have fully and carefully addressed my earlier comments, which were in any case minor.

Reviewer #3 (Remarks to the Author):

In this revised manuscript, the authors do an excellent job of addressing the previous concerns of the reviewers. The new data add clarity to the arguments being made in the paper as well as to the logic of the presentation of how Ncd may cross-link and slide microtubules. The presentation of the material is more robust and clear and better supports the conclusions being made. While I was not in favor of publication of the earlier version of the manuscript, I concur with the conclusions of the other reviewers as well as the strong rebuttal from the authors and support publication of the present manuscript. Some minor comments should be addressed before final publication.

1. The supplemental figure legends are too descriptive of the figure and do not contain sufficient detail regarding what is shown in terms of means, SD, n of sample....
2. The text in the figure part of the model slide is extensive and could be shortened to bring clarity to the figure.

We thank the reviewers for their comments. Addressing all of the comments, we altered the manuscript accordingly. Please find our replies (in blue) to the reviewers' comments (in black) below.

REVIEWERS' COMMENTS:

Reviewer #1 (Remarks to the Author):

The authors have responded fully to my comments and I commend their new experiments.

Reviewer #2 (Remarks to the Author):

i have carefully read the revised ms and the author's responses to my comments and can confirm that they have fully and carefully addressed my earlier comments, which were in any case minor.

Reviewer #3 (Remarks to the Author):

In this revised manuscript, the authors do an excellent job of addressing the previous concerns of the reviewers. The new data add clarity to the arguments being made in the paper as well as to the logic of the presentation of how Ncd may cross-link and slide microtubules. The presentation of the material is more robust and clear and better supports the conclusions being made. While I was not in favor of publication of the earlier version of the manuscript, I concur with the conclusions of the other reviewers as well as the strong rebuttal from the authors and support publication of the present manuscript. Some minor comments should be addressed before final publication.

1. The supplemental figure legends are too descriptive of the figure and do not contain sufficient detail regarding what is shown in terms of means, SD, n of sample....

We added the description (mean \pm SD, *N*) to the Supplementary Figure legends.

2. The text in the figure part of the model slide is extensive and could be shortened to bring clarity to the figure.

We shortened the text part of the Fig. 5 (now Supplementary Fig. 5 due to the editorial request).